Published in FAST Workshop on Smalltalk Related Technologies (11/2022)

# Unicode support in Cuis Smalltalk

**Juan Vuletich**                                                                                   *juan@cuis.st*
*Cuis Smalltalk and LabWare*

**Reviewed on OpenReview:** *https://openreview.net/forum?id=c93fukpVINA*

## Abstract

Any modern development environment needs to provide good support for Unicode text. Cuis Smalltalk presents a specific set of requirements, and offers a unique set of resources for implementing it. Additionally, Unicode has been around for long enough to allow for a critical view on existing implementations. This paper describes the motivations, choices and implementation strategies of the support for Unicode recently added to Cuis.

## 1 Introduction

Unicode (The Unicode Consortium, 1991) is a character set and several encodings for text, covering most of the world's writing systems. Nowadays, software developers expect to be able to write software that can handle user content written in any natural language and alphabet. In addition, there is no good reason not to provide programmers themselves with the same conveniences. Therefore, good Unicode support is a must in any modern programming environment. Of special importance is UTF-8 (Radzivilovsky et al., 2012), the most widely used encoding and part of the Unicode standard. UTF-8 is the recommended and default encoding for text in most platforms and information exchange formats.

Cuis (Vuletich et al., 2022) is an Open Source, multiplatform Smalltalk-80 (Goldberg & Robson, 1983) system. Cuis assumes very little on the underlying system, and this lets it run on many different platforms, using the OpenSmalltalk Virtual Machine (Miranda et al., 2022). Cuis doesn't require or use services from the host platform for most of the features it provides, using instead its own implementations. This includes text encoding, display and editing. This strategy enables full portability of the system and the applications written in it, and it also enables going beyond what the host platform provides, avoiding some of its most irritating limitations.

This paper describes the motivations, choices and implementation of the support for Unicode recently added to Cuis Smalltalk. Its primary audience are Smalltalk developers interested in Cuis. But it is intended also for those working on Unicode support for other Smalltalk dialects or programming systems, and for those interested on revising existing implementations, hoping for improvements.

## 2 Current Landscape

Unicode is a relatively new piece of technology. Not too long ago most programmers would assume that one byte could (should!) be equated to a letter, digit or punctuation mark. This meant that raw bytes were stored in files or memory, and everybody understood they meant essentially ASCII. Soon it became clear that 256 distinct characters is far from enough for writing anything but the most basic content in the small Latin alphabet.

Over time, there were many attempts both to make better use of the limited 8-bit range, and to explore larger character spaces. Later, when the dust started to settle, UTF-8 encoded Unicode emerged as the preferred encoding and character space for most text files and text network traffic (Wikipedia Editors, 2022).

But many programming languages (and their standard libraries) were developed or adapted to handle multilingual text during that time. As a consequence, they adopted the conventions and assumptions that were popular at that moment. Some of these may be:

- The solution is, when appropriate, to use another 8-bit character set, not necessarily the Latin alphabet, and assume that encoding for raw bytes. And dozens of alternative character sets appeared.

- For Asian languages, special multi-byte encodings like Shift JIS, EUC-CN, EUC-KR and EUC-JP.

- A single 16 bit, covering 65536 characters is enough to write in any language. Files should be Little Endian.

- Same as above, but files should be Big Endian.

- Same as above, but files should start with a non-ambiguous mark to state Endianess, that is not characters, and should be removed as soon as the file is read in memory.

- A larger character set, Unicode, is needed. A plain 32 bit encoding is the simplest solution. Each character will take 4 bytes in memory and files.

- A smarter encoding can handle the whole Unicode character space and still be compatible with old 8-bit ASCII files and content. Additionally it has many other good properties, and few drawbacks. This is, of course, UTF-8 encoded Unicode.

Given the difficulties of inventing, implementing, and experimenting with all these possibilities, it is not surprising that the focus was on the technical details. Besides, people came from the idea that everybody was expected to understand the encoding of characters. Most programming languages and their standard libraries made encodings explicit, and required programmers to have a deep understanding of them for their code to work at all. This was OK for ASCII in the '60s, but few realized that the explosion of complexity required a different attitude.

But, for instance, 50 years ago the C language (and even Fortran before that), already helped programmers by hiding the details of the various possible encodings for numeric data, exposing only their external behavior, and automatically selecting the proper algorithms to work on them. Programmers using high level languages (anything but Machine Code and Assembly) were never required to deal with the raw bytes of a float (or int, unsigned int, long int, etc.) number, remembering its encoding, and selecting themselves the proper code to (for instance) add two of them.

It is a sorry state of affairs that, 50 years later, this is now the standard way to deal with plain text.

These problems are exacerbated by the fact that most languages fix these decisions in the language itself, making them impossible to change.

Cuis tries to learn from all these experiences, and to exercise the Smalltalk tradition of providing meaningful abstractions with sensible public protocols, and avoiding accidental complexity. In Smalltalk we also have the advantage that all this belongs in class libraries, not in the language definition. This enables evolution long after the language itself was fixed.

But first, let's take a look at what other programming languages have done. These were chosen because they are representative of the mainstream approaches to the problem. Python 3 is usually considered a good implementation of Unicode. JavaScript has stuck to its original implementation, and is instead considered a rather sloppy one. We'll also comment briefly on other Smalltalk systems.

### 2.1 Issues in the Unicode support in Python

The main difference between Python 3 and Python 2 is a new, incompatible implementation of Unicode. But even if Python 3 is an improvement, it still has poor encapsulation, for example encouraging the view of Byte objects as ASCII strings. The following is verbatim from the official Python 3 documentation (Python Software Foundation, 2022):

*Firstly, the syntax for bytes literals is largely the same as that for string literals, except that a b prefix is added:*

*Single quotes: b'still allows embedded "double" quotes'*

*Double quotes: b"still allows embedded 'single' quotes"*

*Triple quoted: b'''3 single quotes''', b"""3 double quotes"""*

*Only ASCII characters are permitted in bytes literals (regardless of the declared source code encoding). Any binary values over 127 must be entered into bytes literals using the appropriate escape sequence.*

*As with string literals, bytes literals may also use a r prefix to disable processing of escape sequences. See String and Bytes literals for more about the various forms of bytes literal, including supported escape sequences.*

*While bytes literals and representations are based on ASCII text, bytes objects actually behave like immutable sequences of integers, with each value in the sequence restricted such that 0 <= x < 256 (attempts to violate this restriction will trigger ValueError). This is done deliberately to emphasise that while many binary formats include ASCII based elements and can be usefully manipulated with some text-oriented algorithms, this is not generally the case for arbitrary binary data (blindly applying text processing algorithms to binary data formats that are not ASCII compatible will usually lead to data corruption).*

Consider this example of Python code, that takes a String and prints the result of "encoding" it:

```
txt = "My name is Ståle"
print(txt.encode())
print(txt.encode(encoding="ascii",errors="backslashreplace"))
print(txt.encode(encoding="ascii",errors="ignore"))
print(txt.encode(encoding="ascii",errors="namereplace"))
print(txt.encode(encoding="ascii",errors="replace"))
print(txt.encode(encoding="ascii",errors="xmlcharrefreplace"))
```

It generates the following output:

```
b'My name is St\xc3\xa5le'
b'My name is St\\xe5le'
b'My name is Stle'
b'My name is St\\N{LATIN SMALL LETTER A WITH RING ABOVE}le'
b'My name is St?le'
b'My name is Ståle'
```

Note that the answers are, as expected, Byte objects. The weird thing is how those sequences of bytes (not Characters!) show themselves. Consider the last example. We are "encoding" a Unicode String "in ASCII", replacing "errors" with "xml character references". This answers a Bytes object that prints itself nothing like it was asked (i.e. as a sequence of bytes, or perhaps an ASCII string), but exactly as the original Unicode String.

There's no question that to make sense of this, the programmer needs to learn quite a bit about what the language understands for "encoding" and for "Bytes object".

The following pieces of advice are from "Pragmatic Unicode" (Batchelder, 2012).

> *Know what your strings are: you should be able to explain which of your strings are Unicode, which are bytes, and for your byte strings, what encoding they use.*

> *In addition, if you have a byte string, you should know what encoding it is if you ever intend to deal with it as text.*

This is already admitting defeat.

If the system and libraries are to help programmers and not just put the burden on them, we must do much better than this.

## 2.2   Some words about JavaScript

JavaScript is in no way better. The JavaScript String type gives access to internal UTF-16 words, not code points. The official documentation (Mozilla Foundation, 2022) says:

> *The String object's charAt() method returns a new string consisting of the single UTF-16 code unit located at the specified offset into the string.*

Note the name of the method, and the fact that it does not return a character, and that the argument is not an index to a character, but to a 16-bit word. Later, this official documentation tries to help the poor soul learning all this. A screenful of JavaScript code, implementing getWholeCharAndI() is preceded by:

> *Getting whole characters*
>
> *The following provides a means of ensuring that going through a string loop always provides a whole character, even if the string contains characters that are not in the Basic Multi-lingual Plane.*

So, the language includes a method that looks like what a programmer may need (accessing the nth code point in a Unicode String), and it may appear to work, as the programmer exercises their code with multilingual input, including accented Latin letters, Cyrillic, Greek, etc.

But, as soon as someone enters a character not in the Basic Multilingual Plane, the application will produce meaningless garbage. Why would programmers or users need to know or care about the BMP? Why would they need to know that it includes Georgian or Coptic, but not Mathematical Alphanumeric Symbols (that are just styled Latin and Greek letters, and decimal digits), for instance?

ECMAScript 6 introduces the codePointAt() method. This answers, sometimes, a CodePoint. But the argument is, as before, in 16-bit words, not characters. So if you happen to hit the second word of some character not in the BMP, you get instead just that. The second word.

Programmers need to remember to use instead the suggested implementation in the official documentation, that is O(n), needing to iterate the String.

It is easy to imagine that many programmers will fall into these kinds of pitfalls. In addition, the language does automatic conversions between various types, hiding programmer mistakes until much later in the execution, making bugs harder to find.

## 2.3   Not even Smalltalk is perfect

In general, Smalltalk systems that support Unicode do better than Python or JavaScript. Still, in some cases there is poor encapsulation (conflating String and Bytes), or limited compatibility between the various String classes. Some include different encodings that nobody uses, serving only to make the system much harder to understand than necessary.

For example, in Squeak, in system category 'Multilingual-TextConversion', the TextConverter hierarchy has 58 classes, the KeyboardInputInterpreter hierarchy has 25 classes, and the ClipboardInterpreter hierarchy has 17 classes.

With respect to encoding services, Squeak includes both #convertToEncoding: and #convertFromEncoding:. These methods are implemented in String, and the answer is also a String. No ByteArrays involved, even if Squeak includes a ByteArray class.

Strings know nothing about the encoding these bytes may be using. Only for this use, String (actually instances of ByteString) should be assumed to hold bytes and not Characters. But String doesn't know about this, and if such a String is printed or edited, it will appear to contain garbage.

It will now act as if it contained 8-bit Characters, not the bytes of some Unicode string, encoded in UTF-8 or some other of the multitude of possible encodings. In this case it acts just like Python.

## 3   Desiderata

The feeling of dislike caused by the way Unicode was dealt with in other programming languages made this author reflect on the problem, and to try to find a better way for Cuis. The first result of this is an explicit list of desirable features and properites.

In a (Smalltalk) programming environment, good support for Unicode must honor:

- Text editors (and any other UI control or widget that includes text) can handle Unicode, using appropriate outline fonts such as TrueType.

- All source code is Unicode. The whole Unicode range of characters can be used. Code is not limited to ASCII or some other small character set.

- Source code files, and any other text files, are UTF-8.

- System clipboard can exchange Unicode text with other applications.

- The programming language, libraries and environment should not put additional burden on programmers because of Unicode. Most programmers are not deeply interested in the challenges of supporting text written in any script, and that is OK.

## 4   Cuis, before full Unicode support

As Cuis was originally derived from Smalltalk-80 and Squeak, it inherited an 8 bit encoding for Characters and Strings, and a set of bitmap based StrikeFonts. Cuis improved on them, by adopting the ISO 8859-15 Latin Character set, and a set of pre-rendered subpixel antialiased fonts. Later we made good use of the empty area (codes 128 to 159), with a careful selection of mathematical symbols (Notarfrancesco, 2022).

This gave a good user experience, if one is willing to accept a character set covering only the Latin alphabet. Besides, it also had several other limitations that, even if they are widely accepted, as they are imposed by every other GUI out there, they are still arbitrary and should be challenged. The original goal of Cuis Smalltalk was to go beyond these limitations (Vuletich, 2022).

Recently, a new vector graphics rasterization engine, written by the author and based on novel techniques he developed (Vuletich, 2013), became ready for prime time. It was integrated into the OpenSmalltalk VM (OpenSmalltalk.org, 2022) as an internal VM plugin. In parallel, the redesigned Morphic UI framework in Cuis evolved to take full advantage of it. It uses floating point numbers (not integers) for coordinates, and to specify locations of submorphs in an owner it uses affine transformations instead of just displacements.

The result is a Zoomable User Interface (ZUI), where any GUI element may be scaled and rotated arbitrarily. This means that bitmap based StrikeFonts are no longer good enough, as these are pre-built to a specific

pixel size, and only for integer parameters and in an horizontal direction. A text rasterization engine with an unprecedented level of flexibility is needed.

The good news is that the new vector graphics engine in Cuis reaches levels of visual quality and performance previously unseen on portable graphics libraries. This makes it suitable for all text rasterization, and not just for graphics. Now it is possible to rasterize all the text on the GUI in real time, at any non-integer (i.e. Float) position, scale and angle, using TrueType fonts. This is of course in line with the traditional ethos of Smalltalk (Ingalls, 1981).

This new approach to text rasterization has another important consequence, besides enabling a high quality ZUI. It frees us from the need to keep pre-rendered bitmap fonts at all desired pixel sizes in memory. This substantial memory saving allows loading large TrueType fonts in memory. Unicode support is now within reach.

## 5   General criteria, for meeting Desiderata

Let's specify in more detail the features that we require from the Unicode support in the system. The following list of criteria starts with common sense reflection on Unicode and on the Smalltalk system, but later turns more into a reaction to the mistakes done in other languages, especially Python and JavaScript.

- Characters from all scripts are on equal footing. ASCII characters should enjoy no special privileges.

- Comments in source code can include any Unicode character.

- String literals in source code can include any Unicode, without the need for escape sequences or any other special syntax.

- Binary selectors are mathematical symbols. Any mathematical symbol can be used for binary selectors, and not just those included in the ASCII character set.

- Although we prefer to use only English in source code, names used in source code can include letters from any alphabet, if desired. This includes class and global names, method unary and keyword selectors, and names of variables of all kinds.

- The main String and Character classes must support Unicode. Any other String or Character class with a more limited range is fully compatible with them, and their instances are automatically converted as needed. Programmers don't need to be aware of them. This is polymorphism in action, just like in the Number hierarchy.

- The classes provided to the programmer should represent a meaningful abstraction. Not more, not less. This sounds obvious but isn't the case in many implementations. In particular, String instances are not sequences of bytes: they have no byte access protocol and they can't be 'decoded'. They are sequences of characters, and have character access protocol. Their internal representation, using any encoding as needed, is irrelevant to most programmers, and protected by encapsulation. Byte sequences are held in ByteArrays. ByteArrays are not Strings and are not automatically converted into them.

- Application code is not concerned with text encoding. Classes such as UniFileStream and Clipboard deal with them as needed.

- Existing code packages that don't do low level String manipulation should not be affected. The migration of existing package files, from ISO 8859-15 to UTF-8 should be transparent.

- Behavior of code should be deterministic. The risk of application code working on a system and breaking on another because of a difference in default encoding or locale can not be accepted.

- The base system doesn't need to support dozens of legacy encodings, making them a central part. The few developers who need them can use optional packages that supply them. These could provide, for example, services to decode ByteArrays (never Strings!) using various encodings.

# 6   Subprojects

Several distinct pieces of functionality needed to be developed to meet these criteria. The following sections describe them and their dependencies.

## 6.1   UTF-8 String and Symbols, Unicode Characters

In order to avoid the expensive encoding and decoding of Unicode CodePoints as much as possible, Cuis uses UTF-8 as its internal encoding for Unicode Strings. This avoids both operations when using the universally accepted UTF-8 files. Thus, no decoding is needed to display UTF-8 file contents since as previously stated, it is possible to design a very fast and efficient data structure that is directly accessed with the UTF-8 bytes.

The central class is Utf8String. It offers $O(1)$ access time to the stored CodePoints and is based on (Van Caekenberghe, 2022). Instances are conceptually immutable: #at:put: is not implemented, as it can't be done in $O(1)$: in many cases the UTF-8 byte sequence would need to be recreated.

The existing String class, that can represent 8 bit Characters iskept. Utf8String shares a new common superclass with the existing it, called CharacterSequence. Over half of the code of String was moved to this common superclass, making String and Utf8String focus on their specific details.

String and Utf8String are fully polymorphic and their instances are interchangeable. This means, for instance, that if one instance of String and one of Utf8String have the same characters, they will answer true to #=, and they will have the same #hash value. To make this efficient, a 'hash' instance variable was added to Utf8String. It is reasonable to do this only because Utf8Strings are essentially inmutable, and therefore the hash value can not change.

Utf8String doesn't have byte access protocol. This is important. To store bytes we use ByteArrays, that are not Strings. Just keeping these separate avoids most of the confusion and bugs people face when working with Unicode.

But making Utf8String immutable required modifying any code in the system that would attempt to modify existing String objects. Surprisingly, this happened in just a handful of places, and the changes were relatively easy. Most of them just created new instances of String and filled them with Characters. The updated code calls #streamContents: or #writeStream, that are implemented in Utf8String class using Utf8EncodedWriteStream, a new class that is used only for creating instances of Utf8String.

For compatibility with existing code that creates WriteStreams on Strings, in case an Utf8String is passed, it is automatically converted into an ArrayOfCharactersAndCodePoints. This is transparent: when done streaming, #contents will correctly answer an instance of Utf8String.

Class Utf8Symbol is subclass of Utf8String. Instances of Utf8Symbol, Utf8String, String and Symbol having the same characters will answer true to #=, and will have the same #hash value.

As Symbols are meant to be uniquely created, there is a single Symbol table that holds both instances of Symbol and Utf8Symbol. When writing regular Smalltalk code, the programmer doesn't need to know if some object is an instance of the old String / Symbol or the new Utf8String / Utf8Symbol.

Thanks to a quick #isAscii test, most operations are usually as fast as in String: most of the comparisons and conversions can be done using existing primitives. Same for concatenation and many other operations.

To represent Unicode Characters, a new class UnicodeCodePoint was added. It is fully polymorphic with Character. Their instances are interchangeable. The General Category in the official Unicode Character Database (The Unicode Consortium, 1999) is used to determine if a UnicodeCodePoint is an uppercase or lowercase letter, or a symbol valid in a binary selector, or neither of these.

## 6.2 TrueType fonts

In order to support arbitrary geometry transformation with high quality rasterization, Cuis needs to hold in memory the outline information for all glyphs in the required TrueType fonts. Additionally, it needs to render them in real time, without caching pre-rendered glyphs.

A representation using a single Array of glyphs for each font is very inefficient, since it would be mostly empty. This i because most TrueType fonts only cover small subsets of the whole Unicode range of possible CodePoints. Besides, it would require converting from UTF-8 bytes to CodePoints each time a glyph needs to be accessed, for example, to draw it on the screen.

A less naive design could make good use of the fact that text will be a stream of UTF-8 bytes: a first Array of size 256 could be indexed with the next byte in the stream. The retrieved object could be an instance of Glyph (for 1-byte CodePoints), or another Array. This second Array would be indexed with the next byte possibly finding yet another Array, and so on, until an instance of Glyph is found. This Glyph object contains enough information to render the Character on a Display, including glyph metrics and contours. This process would continue with the next byte and so on. One problem with this approach is that text display needs to be fast, and even if using primitive code, a rather expensive traversal of Smalltalk objects would be needed. Additionally, the number of Smalltalk objects needed to represent such structure would be very large, imposing a serious overhead on the Garbage Collector.

Cuis uses a different approach. Each TrueType font uses just two large Arrays. The first one is a FloatArray holding all the relevant geometric parameters that describe each glyph and the Bezier curves to draw them, one after another. The other one is an IntegerArray, indexed using the bytes of the UTF-8 encoding of a CodePoint. For each byte, except for the last byte of some CodePoint, the integer found is an offset to be added to the next byte. This number is used to index again the very same IntegerArray, to continue the traversal. For the last byte of the UTF-8 encoding of the code point, the number found is the index for the glyph parameters in the FloatArray. To make these two cases easy to distinguish, in the latter case the index is multiplied by -1 (i.e. it is stored as a negative number). This reduces the number of objects allocated to essentially two, even for very large TrueType fonts. As these two Arrays are Words objects, not Pointer objects, the work of the Garbage Collector is greatly reduced. Additionally, the primitive code that needs to use them is faster and easier to write.

To read .ttf files, Cuis uses a TrueType font reader inherited from Squeak, written by Andreas Raab. It creates instances of a Smalltalk model of the glyphs, that is close to the .ttf file format. These Glyph objects are later converted into the format previously described, and discarded.

## 6.3 UTF-8 Text Display

Cuis includes a vector graphics rasterization engine. There are two versions of it. One is written in pure Smalltalk, and is meant for learning, experimentation, and development of new features. The other one is a virtual machine plugin, and it is part of the official OpenSmalltalk VM (OpenSmalltalk.org, 2022). This graphics backend can draw high quality Bezier curves and other vector primitives. It uses float (not integer) coordinates and geometry transformations, and it has sub pixel resolution. It uses the Signal Processing approach to anti-aliasing (Vuletich, 2013), resulting in both higher quality and performance than standard libraries such as Cairo (The Cairo Developers, 2003).

For Unicode text display, the vector graphics engine was augmented with new primitives, that iterate over a UTF-8 byte sequence, traversing the TrueType font data structures as previously described. For each glyph, geometry parameters are retrieved from the FloatArray. Next, the parameters for the Bezier curves that comprise the glyph contours are read, and the Bezier curve primitive is used to draw the glyph. The position for the next glyph is updated with this glyph's advance width, and iteration of UTF-8 bytes continues with the following glyph.

Additionally, Cuis implements optional caching of pre-rendered glyphs, although only if there is no rotation or scaling. This is useful for large quantities of text and programming tools, especially on slow hardware. The downside is that position for cached glyphs will be rounded to integer pixel coordinates.

### 6.4 Unicode enabled text editors

The text editors in Cuis were adapted to work equally well on instances of Utf8String or String. The changes were minor. It was just needed to remove direct references to the String class, using a Preference to select the preferred String class, and converting the initial contents to it.

Additionally it was needed to update any code that attempts to modify existing String instances, as Utf8Strings are immutable. This doesn't happen often: as Strings were already of a fixed size, set at creation, most editing operations already needed to create a new instance. It was just a matter of using the required way of doing this, using #streamContents: that will do the correct thing.

Text editors take advantage of UTF-8 text display using TrueType fonts, simply by asking the contents to draw themselves.

### 6.5 FileStreams

A new UniFileStream class replaces the older StandardFileStream. Like its predecessor, it can be in several modes, that determine how file contents are interpreted when read. But instead of just 'text' and 'binary', now modes are #useBytes (for binary data), #userCharacters (for ISO 8859-15 Characters) and #useUtf8String for UTF-8 encoded Unicode, actually the default mode. Invalid UTF-8 byte sequences will be interpreted as ISO8859-15 Characters, allowing for seamless compatibility with legacy files. For writing, the mode is not used, and the saved bytes depend only on the object being written. Numbers are assumed to be bytes. Characters and Strings of all kinds are saved as UTF-8.

As previously stated, it is assumed that text files are encoded in UTF-8. This means that the base system and most application developers don't need to deal with files using any other Unicode encodings. UniFileStream is therefore relieved of that extra complexity. The need to deal with other encodings is considered part of the problem domain of specialized applications.

Note that #position and #position: assume positions are expressed in bytes, as it is usual with files. This means that computing positions in client code is not recommended, but asking a #position to be used later with #position: to go back to the same place is safe. To traverse files in the default #useUtf8String mode, messages such as #next, #peek, #peekLast, #skip and #skipBack all advance or go back one CodePoint at a time. In the same vein, #next: takes the number of CodePoints to advance and answer, and #skip: take the number of CodePoints to advance (or recede if negative). These are the preferred ways to traverse files, without client code needing to take care of the variable length property of UTF-8 encoding.

### 6.6 Smalltalk Parser

The Smalltalk parser works both on ReadStreams and file based Streams, i.e. UniFileStreams. It used to make the assumption that doing #next would advance #position by one, and that it was OK to do stream position arithmetic itself. But while this is indeed true for ReadStreams on Utf8Strings, it is no longer true for UniFileStreams. The code that did this had to be rewritten to use Stream protocol to advance or recede, and only call #position: with values previously obtained by calling #position. Fortunately, the result is cleaner code, easier to understand and maintain.

Another improvement that was needed in parser was to replace explicit lists of, for instance, valid characters in binary selectors, with calls to #isValidInBinarySelectors. The result is again cleaner, more general code.

In most Smalltalk implementations, Characters are never duplicated. The Parser made heavy use of this assumption, using #== to compare Characters, especially an unknown Character and a literal Character. But with UnicodeCodePoint this assumption is no longer true. Even if UnicodeCodePoint were made unique, immediate objects, the same ASCII character could exist both as a Character and as a UnicodeCodePoint. All these calls for #== needed to be found and replaced with #= .

### 6.7 UTF-8 Smalltalk code

The previously described features makes it now possible to use Unicode anywhere in Smalltalk code.

Thanks to the good properties of the UTF-8 encoding, existing code files using non-ASCII Latin characters (i.e. using the older ISO 8859-15 support in Cuis) are almost always invalid if interpreted as UTF-8 encoded Unicode. This means that such code will be read, interpreted in the old encoding, and converted to the correct Utf8String. The programmer can edit such code at will, and then it will get saved in UTF-8. Simply by editing and saving existing methods, the user's packages and change sets will be converted to the new standard.

Even if a programmer doesn't care about these details, their code will become easier to handle in external tools, or be ported to other Smalltalk systems.

## 7  Cleanup

The development strategy was incremental, and assumed that the old behavior was always available to implement the new features. But if these can be assumed to be present, and the base system can rely on them, it is possible to forego some legacy and interim solutions, now made obsolete. We can remove:

- StrikeFonts and their display mechanisms. This will free TrueType to evolve on its own, without being limited to what StrikeFonts could do.

- ISO 8859-15 in Characters and Strings, and support for it in TrueType. It still makes sense to keep ASCII only Characters and Strings, though.

- BitBltCanvas. It provides Morphic drawing services using BitBlt. It is replaced by VectorCanvas, that uses the new vector graphics engine.

- HybridCanvas. This class handled the drawing of Morphs, forwarding to the old BitBltCanvas as much possible, allowing for a usable system even without the VM plugin.

- FileStream and StandardFileStream. They are replaced by UniFileStream.

This will most likely be done in the upcoming Cuis 7 release.

## 8  Future work

As previously said, Unicode is large and complex. In the initial implementation for Cuis the focus has been on a healthy approach with good qualities for the benefit of application developers. Many advanced features of Unicode were left for future development. This first release of Unicode support in Cuis has thus the following known limitations. Removing them should be seen as extensions to the work presented here.

- Collation is only correct for the Latin alphabet.

- Combining characters are not handled.

- Character equivalence, composition, decomposition and normalization are not yet supported.

- Explicit support for Grapheme Clusters will be added in the future.

- No support for bidirectional text.

- No ligatures.

There is indeed room for future improvement.

## 9   Conclusions

This paper presents the Unicode support added to Cuis Smalltalk. We identify problems in Unicode support as implemented in other programming languages and their libraries, and we make the case that a programming language should aid programmers, relieving them of accidental complexity and irrelevant technical details. It should help them produce code that has few bugs, and is easy to understand and modify. The goal here is to make programming with Unicode not much harder than programming with plain ASCII.

We identify a set of desirable properties and general criteria to be followed. We describe the Cuis implementation of Unicode, and how it attempts to address these needs. While there are advanced Unicode features that are not yet implemented, those fundamental criteria have been fulfilled, providing a solid foundation for the development of applications and for future development of Cuis itself.

Finally, we encourage readers to experiment with the system, and to engage in public discussion on how to keep improving.

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
