# OpenReview forum: "Unicode support in Cuis Smalltalk"
_FAST.org.ar/2022/Workshop — FAST Smalltalk 2022_

### Official Review · Reviewer_oHmo · 2022-10-20
**Interesting and Important**

**Rating:** 8
**Confidence:** 5

**Review:**

I would like to first thank Juan for this write up.
Understanding the pitfalls of text (strings) and encodings is a big problem for most programmers, and unfortunately I don't see it taught in most curricula in universities...

# Paper Summary

The paper presents the problems in String design in languages such as Python, Javascript, and even Squeak's Smalltalk dialect.
Then it outlines a list of desired properties, before diving into the several redesigns inside Cuis' libraries including the parser, the text editor, the font rendering system.

# Comments

I find the paper interesting and relevant for today's common knowledge on encodings.
I cannot say how many new and old developers I meet that still assume ascii as granted.

From the entire writing, only one thing escaped to me. The paper is written from the internal Cuis perspective, where developers should not care about the encodings of strings, and that's ok. The decision to make strings utf8 encoded internally seems a rather sensible one that can improve performance in default cases. However, it is not clear to me what strategy was chosen to interact with other encodings. For example, Pharo has as a general design read and write streams that are configured respectively with a decoder or encoder. It is not clear whether UniFileStream serves this purpose too.

Finally, I had while reading the paper the impression that the paper targeted a knowledgeable audience, that is already versed in the matter of encodings. If I may propose an enhancement to make the paper more open to another target and self-contained, I believe the paper could use an more functional introduction to encodings in addition to the more historical one. At least, I did not notice the dichotomy between Strings and bytes be emphasized until late in the paper.

---

### Official Review · Reviewer_jtyE · 2022-11-01
**Implementation focuses, structure could be improved**

**Rating:** 6
**Confidence:** 4

**Review:**

The paper gives an overview of the implementation of Unicode support in Cuis Smalltalk.

It first presents Unicode support in some other systems (Python, javascript, other Smalltalks). The paper then discusses then desiderata for Cuis. The implementation is described (with all sub projects that are affected). The paper finishes with a short discussion of future work.

Points to improve:

- The paper could benefit from a quick overview of Unicode. For example, the term "Codepoint" is mentioned the first time while discussing Javascript (section 2.2). What is it? How is this handled for the other systems? It seems to be important concept.

- What about other languages? Is there a reason to just discuss python, javascript and Smalltalk?

- The desiderata seem to be a reaction to the mistakes of the other implementations (see 5 "later turns more into a reaction to the mistakes done in other languages"), but the relationship to the prior sections
is not explicit.

- could every language section  have a clear list of mistakes (and good properties) so that Section 5 is just a summary?
- It is not clear how these relate to the other systems. Are some of these already solved by the prior discussed implementations?

Structure of Section 6.
- Would it not make sense to start with the core part (which is now 6.3)?
- It misses a comparison to the other systems
- it does not get clear how the criteria in 5 drive the implementation decisions

- The end of the paper (after Section 6), it would be interesting to discuss if the criteria have been fulfilled.
- How does the future work relate to the criteria? Are any not fulfilled yet?